# Insufficient Cold Resistance as a Possible Reason for the Absence of Darkling Beetles (Coleoptera, Tenebrionidae) in Pleistocene Sediments of Siberia

**DOI:** 10.3390/insects15010064

**Published:** 2024-01-16

**Authors:** Roman Yu. Dudko, Arcady V. Alfimov, Anna A. Gurina, Ekaterina N. Meshcheryakova, Sergei V. Reshetnikov, Andrei A. Legalov, Daniil I. Berman

**Affiliations:** 1Institute of Systematics and Ecology of Animals, SB RAS, Novosibirsk 630091, Russia; rdudko@mail.ru (R.Y.D.); auri.na@mail.ru (A.A.G.); reshetnikov-art@yandex.ru (S.V.R.); 2Institute of Biological Problems of the North, FEB RAS, Magadan 685000, Russia; arcalfimov@gmail.com (A.V.A.); dberman@mail.ru (D.I.B.); 3Institute of Biology, Ecology and Natural Resources, Kemerovo State University, Kemerovo 650000, Russia; 4Department of Ecology, Biochemistry and Biotechnology, Altai State University, Barnaul 656049, Russia; 5Biological Institute, Tomsk State University, Tomsk 634050, Russia

**Keywords:** cold resistance, supercooling point, low lethal temperature, Tenebrionidae, Chuya Depression, Altai Mountains, Central Asia, late Pleistocene, tundra-steppe

## Abstract

**Simple Summary:**

One of the main differences between the modern fauna of steppe and desert-steppe insects and the similar fauna of the Last Ice Age is the current leading position of darkling beetles (Coleoptera, Tenebrionidae), which are represented in fossil faunas only by singletons. We hypothesize that the reason lies in the insufficient cold resistance of these insects for successful overwintering. We studied the cold resistance of adults from five species of darkling beetles from the Altai Mountains that overwinter in the soil and the larvae from one such species. The ranges of three of these are limited from the north by the desert-steppe margin of Central Asia with extremely low air temperatures. More than 50% of individuals of these species in the experiment did not withstand cooling below −22 °C. Temperatures in the soil of natural habitats at a depth of 10 cm are close to or lower than the above values. Overwintering is therefore possible in places with greater snow thickness (hollows, wind shade of shrubs and large cereals). Since darkling beetles are now on the border of temperature resistance in the Altai, they likely did not exist in the much more severe conditions of the glacial periods in the Altai Mountains, West Siberian Plain, and Northeast Asia.

**Abstract:**

The level of diversity and abundance of darkling beetles (Coleoptera, Tenebrionidae) is the main difference between the late Pleistocene and modern insect faunas of arid regions. In the Pleistocene assemblages they are extremely rare, whereas in the modern ones they predominate. It is assumed that the reason for their rarity in fossil entomological complexes is their lack of cold resistance. The supercooling points (*SCP*) and low lethal temperatures (*LLT*) of adults from five species of Altai darkling beetles that overwinter in the soil and larvae from one such species were measured in the laboratory. All beetles supercooled at negative temperatures but could not survive freezing, with the average *SCP* of the most cold-resistant species between −25.7 and −21.7 °C (*Bioramix picipes*, *Anatolica dashidorzsi*, and *Penthicus altaicus*). However, 50% of the individuals from different species in the experiment died after exposure during two days at temperatures ranging from −22 to −20 °C. The focal species are distributed in parts of Central Asia with an extreme continental climate, and the temperatures measured in the soil of these natural areas turned out to be lower than or close to the limit of cold resistance of the beetles. Overwintering of darkling beetles is therefore only possible in areas with deep snow: in hollows, under bushes, and under large cereals. Darkling beetles with poor cold resistance could not have existed in the colder climate of the late Pleistocene, which explains their absence from fossil fauna.

## 1. Introduction

Possible predictions for the currently observed rates of climate change can probably be “read” in the deposits of the late Pleistocene and Holocene epochs, known for significant climatic fluctuations. The fossil remains of insects that still live today are known to be the most sensitive indicators of the parameters of former environments [1]. Their study has significantly advanced the understanding of the paleogeographical environment of geological times in many regions of the world [1,2,3,4,5].

One of the striking features of Pleistocene entomocomplexes (“faunas”) is the almost complete absence of darkling beetles (Tenebrionidae, Coleoptera), which are now characteristic of arid regions. Meanwhile, in Northeast Asia, during the cryoarid episodes of the Pleistocene, starting from its early stages, steppe insects of many groups, primarily beetles of other families and hemipterans, were an essential component of fossil faunas; about 20 similar species were found here [4]. They are classified as steppe inhabitants since they are currently found mainly in the steppe zone and/or in the mountain steppes of Southern Siberia. Currently, in tiny areas of steppe vegetation remaining in the upper reaches of the Indigirka and Kolyma rivers, at least 54 similar species of beetles and true bugs have been identified; darkling beetles are not found either in fossil faunas or in the modern faunas of Northeast Asia [6].

An even greater surprise appeared to be the very poor representation of darkling beetles in the late-Pleistocene deposits of the south-west Siberian Plain. A rich fauna of Coleoptera (more than 400 species) has been identified here, in many ways similar to the modern fauna of the steppe depressions of Southern Siberia [7,8,9].

Meanwhile, darkling beetles are one of the largest families of Coleoptera: there are more than 19,000 species in the world fauna, and more than 4500 known in the Palaearctic fauna [10]. They are one of the dominant groups in arid regions; in the intermontane depressions of Southern Siberia and Mongolia, they predominate numerically among ground-dwelling animals of the medium size class [11,12]. For example, there are 63 species of darkling beetles in the Uvs Nuur Depression, inhabiting almost all landscapes—from desert-steppe to forest on the slopes of the Tannu-Ola Mt. Range [11].

It would seem that a plausible explanation for the absence of darkling beetles in Pleistocene entomocomplexes and now in relict tundra-steppe groups of Northeast Asia may be associated with severe climate conditions. However, we are not aware of any specific studies of this kind, and isolated publications on the cold resistance of darkling beetles leave wide scope for interpretations.

In the arid regions of Central Asia (Gurbantonggut Desert), the winter-cold resistance of *Microdera punctipennis*, which tolerates short-term exposure to −20 °C, was studied [13]. The minimum temperature that five species of darkling beetles from South Africa can survive is much higher, ranging from −7.1 to −8.9 °C [14]. Only the dendrobiont *Upis ceramboides* (Linnaeus, 1758) tolerates freezing below −83 °C [15,16].

Our goal was to evaluate the role of the cold resistance of several species of darkling beetles from cold arid regions in limiting their ranges and, if possible, to use the results to elucidate reasons for the absence of darkling beetles in Pleistocene fossil remains.

To achieve these goals, the following four tasks had to be completed.

Measurement of the cold-resistance parameters of several species of darkling beetles from the arid highlands of Altai (Chuya Depression), a region known for its cold winters.Based on the literature, assessment of the climatic (air and soil temperatures, snowdepth) conditions in the modern range of the darkling beetles studied.Estimation of specific temperature conditions in the soils of the darkling beetle habitat using data loggers.Correlation of the cold resistance of darkling beetles with temperature conditions in the modern range of the studied species in the South-Eastern Altai and Central Asia, and comparison to the reconstructed conditions of the tundra-steppe territories of Western Siberia and the Northeast Asia in the late Pleistocene.

## 2. Materials and Methods

### 2.1. Region and Location of Work

The choice of region and location for catching darkling beetles was based on the following principles:It should be an arid region with extremely low temperatures in winter.The entomofauna of the region should be close to the Pleistocene fauna of the West Siberian Plain.The region should be located near an operating weather station in order to be tied to a long-term climate base.

Complexes of Coleoptera from Pleistocene deposits in the south of Western Siberia are most similar to the modern faunas of South-Eastern Altai [7,8]. The region has a hyper-continental climate, with minimum winter temperatures reaching −55.1 °C in the Chuya Depression [17,18]. Therefore, this particular sampling region was chosen.

### 2.2. Selection of Darkling Beetle Species

The following requirements were imposed on the selection of darkling beetle species to determine cold resistance:They should be abundant, with autumn activity.Species known from Pleistocene deposits are desirable.If possible, species should belong to different taxonomic groups to reveal patterns across different phyletic lineages of the family.

On 17–18 August 2022, in the Chuya Depression (Kosh-Agach Village) and on the adjacent Kuraisky Mt. Range from the north (in the area of Tabozhok Mt.) and the Southern Chuysky Mt. Range from the south (Beltir village) (Figure 1), samples were collected from three species of darkling beetles that were numerous at the time: *Bioramix picipes*, *Anatolica dashidorzsi,* and *Penthicus altaicus*. On August 16, in the adjacent region (Central Altai) in the Ongudai Village area (Figure 1), *Pedinus femoralis* was collected; it is represented by singletons in late Pleistocene deposits (marine isotope stage 3) of the Southern Ural foreland [9]. *Blaps lethifera,* which lives together with *Pedinus femoralis*, was studied as a control. Low resistance to cold was expected for this species associated with mammalian burrows [19].

Beetles were collected in the daytime from under shelters (stones, dung) and in grass turfs. In addition, beetles that were nocturnally active were collected using LED lights. In the first centimeters of soil and under small stones, 7 individuals of *Bioramix picipes* larvae were also found. Thus, the experiment involved adults of 5 species of darkling beetles from 5 tribes of 2 subfamilies and larvae of one species (Table 1).

The darkling beetles were transported in a thermostable cooler box (5–8 °C) to the Institute of Biological Problems of the North, Far Eastern Branch of the Russian Academy of Sciences (Magadan), which has the necessary equipment for cold-resistance assays.

### 2.3. Distribution and Ecology of Species

Subfamily Pimeliinae

Tribe Tentyriini

*Anatolica dashidorzsi* Kaszab, 1965 (Figure 2a)—north-central Asian species. Inhabits dry habitats from saline meadows to rocky steppes and *Nanophyton* deserts in the intermontane depressions—Uvs Nuur, Üüreg Nuur and Chuya [12,20]. The species is geographically variable, so four poorly differentiated subspecies are distinguished. *Anatolica d. subalpina* Kaszab et Knor, 1976 ranges South-Eastern Altai (Figure 2b). Species of the genus *Anatolica* overwinter in the larval and adult stages, the lifespan of an imago is about a year [11].

Subfamily Tenebrioninae

Tribe Blaptini

*Blaps lethifera* (Marsham, 1802) (Figure 3a)—widespread in the Western Palearctic from North Africa, Turkey, Iran, Middle Asia, and Kazakhstan in the south to Scandinavia and the Baltic states in the north, and from Great Britain in the west to the south of Western Siberia in the east [21]. The species inhabits forest belts, steppes, fields, and pastures. Imago, pupae, and larvae overwinter in rodent burrows [19].

Tribe Pedinini

*Pedinus femoralis* (Linnaeus, 1767) (Figure 3b)— distributed in the forest-steppe and steppe zones of Europe, Western Siberia, and Northern Kazakhstan. Common in the steppe depressions of Central Altai up to the Kurai Depression in the east [12]. The species overwinters as imago and larvae of various ages. The lifespan of an imago is 1.5 years [19].

Tribe Opatrini

*Penthicus altaicus* (Gebler, 1829)—north-central Asian species (Figure 4) that is one of the most eurytopic species of darkling beetles in the region. Inhabits the bottom of intermontane depressions to the forest belt, but prefers steppe slopes with gravelly soil [11,22].

Tribe Platyscelidini

*Bioramix picipes* (Gebler, 1833) (Figure 5a)—distributed in the south of Siberia, Transbaikalia, Tuva, and Mongolia (Figure 5b). Inhabits mountain steppes and forest-steppes, where dryness and heat regularly alternate with moisture and coolness [22]. According to our observations, this is one of the least xerophilic species of darkling beetles in the South-Eastern Altai. At altitudes of 2200–2300 m, it is found as numerous in sparse larch forests; higher up (up to 2500 m) it prefers forb or shrub steppes.

Based on their distribution and habitat characteristics, the above-mentioned species belong to two groups. Species of the first one (*Blaps lethifera*, *Pedinus femoralis*) are widespread in the temperate continental and continental climates of the forest-steppe and steppe zones, reaching Central Altai in the east. The second group includes *Bioramix picipes*, *Anatolica dashidorzsi*, and *Penthicus altaicus*—species inhabiting the intermontane depressions of Southern Siberia and Mongolia with a hyper-continental climate. Moreover, typical habitats for *A. dashidorzsi* and *P. altaicus* are rocky desert steppes, while the mesoxerophilic *Bioramix picipes* prefers the somewhat milder conditions of the steppe and forest-steppe type on mountain slopes.

### 2.4. Studying the Resistance of Darkling Beetles to Negative Temperatures

To determine the resistance of darkling beetles to low temperatures, insects collected in August were prepared for overwintering through a process of cold acclimation (see below). After this, two parameters were measured. The first was the supercooling point (*SCP*), i.e., the temperature at which spontaneous freezing begins in a supercooled liquid. If animals die after cooling below SCP, this means that they cannot tolerate freezing (they are classified as “freeze-intolerant”) and overwinter in a supercooled state. The second parameter was the low lethal temperature (*LLT*), i.e., resistance to prolonged exposure to cold. Cold resistance was determined in adults of five species and larvae of *Bioramix picipes* (Table 2).

#### 2.4.1. Insect Acclimation

Darkling beetles were placed in groups of 10–40 individuals in plastic containers with holes for ventilation and a coconut fiber substrate. To select the regime of autumn acclimation of beetles in the experiment, we used previously obtained data on the seasonal variation of soil temperatures in darkling beetle habitats. Since temperature dynamics in microhabitats varied, this study used average values to smooth out sharp changes (Figure 6).

Darkling beetles of all species were acclimated in the temperature range from 10 to −3 °C according to a single scheme, which lasted a total of 76 days (Figure 5). At temperatures of 10 and 5 °C, insects were kept in cooling thermostats TSO-1/80 SPU (Russia). At temperatures from 3 to −3 °C, they were kept in a MIR-254 thermostat (PHC Holdings Corporation PHCbi, Tokyo, Japan).

To clarify the effects of acclimation, *SCP* was determined twice in *P. altaicus*: in the autumn before acclimation (from 27 August to 2 September) and after its completion (from 18 November to 23 November). A temperature of −5 °C (“mild overwintering”) was taken as comfortable for wintering, being on average 2–9 °C higher than temperatures observed in darkling beetle habitats during this period.

#### 2.4.2. Determination of the Supercooling Point (SCP)

*SCP* was measured in a programmable test chamber WT 64/75 (Weiss Umwelttechnik GmbH, Hamburg, Germany) using manganin–constantan thermocouples (wire diameter 0.12 mm). The signal from the thermocouple was recorded by a computer. To ensure uniform cooling, thermocouples were installed in groups of 3 pcs. in a closable thick-walled (2 mm) copper box (5 cm × 8 cm × 8 cm) on thin foam plastic, which was placed in the test chamber. The rate of temperature decrease when determining *SCP* was 0.1 °C per minute based on similar experiments [26,27]. To determine *SCP*, insects were extracted from the frozen substrate and, one at a time, attached to the thermocouple junction with Vaseline.

#### 2.4.3. Determination of Low Lethal Temperature (LLT)

After completion of a mild overwintering acclimation period, the survival of imago samples was determined after exposure to various temperatures in the WT 64/75 test chamber. The choice of *LLT* temperatures was based on the obtained *SCP* values: (1) testing began with a value equal to the average *SCP*; (2) to determine freezing tolerance, insects were cooled down below the lowest *SCP* value for the species.

To control possible temperature gradients in the chamber volume, a calibrated temperature data logger DS1922L (Elin, Moscow, Russia) was placed in each container with insects. The individual error of the loggers calibrated at 0 °C was +0.2 °C (the error declared by the manufacturer was ±0.5 °C).

The temperature in the chamber was decreased by 0.5 °C per hour. The darkling beetles were kept at each of the tested temperatures for 2 days, after which they were heated to 0 °C at a rate of 1 °C per hour, and then to 5 °C in 1 °C increments, keeping them for one day at each temperature. The insects were then transferred to 10 °C for 12 h. At 10 °C, darkling beetles were awakened to determine if they began to move. The insects were observed for two weeks; individuals that restored typical behavioral reactions were considered survivors.

### 2.5. Study of Overwintering Conditions

#### 2.5.1. Winter Temperatures in the Range of the Studied Species

Analysis of the minimum soil temperatures at different depths in the range of the studied darkling beetle species is difficult due to the lack of published data for Mongolia, China and the few weather stations in Russia. A possible way to estimate the temporal and spatial variability of this parameter is to calculate it for a depth of 3 cm using the method of A.M. Shulgin [28]. It is based on weather station data on snow depth and minimum air temperatures. It is clear that temperatures at a depth of 3 cm are not of interest in connection with the wintering of darkling beetles, since the insects are not located this close to the surface. However, based on the diagram for temperatures at 3 cm and using data from the data loggers and weather stations using soil thermometers, one can get an idea of the stratification of temperatures in soils. Thanks to the availability of the initial data for calculations, it is possible to judge the temporal and spatial variability of wintering conditions for insects over a vast territory.

Required data for 1950–2020 was obtained from the electronic databases of VNIIGMII-MCD and the National Centers for Environmental Information [29], as well as from climate reference books [30,31,32,33,34].

#### 2.5.2. Overwintering Conditions for Darkling Beetles in Natural Habitats

To assess the overwintering conditions of darkling beetles in natural habitats, soil temperatures at a depth of 1 and 10 cm and air temperatures at a height of 1.5 m were measured from autumn to spring at intervals of 4 h using TR-2V data loggers (Engineering Technologies LLC, Russia); measurement accuracy was ±0.6 °C in the range from −20 to −10 °C and ±0.5 °C from −10 to 65 °C. The loggers were installed on the bottom of the Chuya Depression, 5 km from the Kosh-Agach Village, as well as on the southern macroslope of the Kuraisky Mt. Range, near Mt. Tabozhok. Temperature measurements were carried out during the 2021–2022 and 2022–2023 seasons (Table 3).

To confirm success of the overwintering of darkling beetles of the South-Eastern Altai near the data loggers, as well as to establish beetle overwintering depth and microhabitats, observations were carried out in early spring (8 April 2023) before the start of their activity.

## 3. Results

### 3.1. Cold Resistance of Darkling Beetles

Insects of all tested species, except *B. picipes*, burrowed singly into the substrate in the laboratory when the temperature dropped, without creating hibernaculae. *Bioramix picipes* formed chambers 1–2 mm larger than the 1–3 individuals filling them; when frozen, their walls became covered with hoarfrost.

#### 3.1.1. Supercooling Point

Under conditions simulating mild overwintering conditions (−5 °C), adults and larvae were motionless, did not respond to touch, and were soft. Distributions of the *SCP* of imago of all species were normal. The variability for all species was large, except for *Blaps lethifera*, whose temperature variation is small, about 5 °C (Table 4).

The studied species were divided into three groups based on the similarity of average *SCP* values (Table 4): the first included *Penthicus altaicus* and *Anatolica dashidorzsi*, where the mean *SCP* did not differ (*F*_1,76_ = 0.37, *p* = 0.55); the second group includes *Bioramix picipes* and *Pedinus femoralis*, where the means were also statistically indistinguishable (*F*_1,52_ = 3.1, *p* = 0.08); the third group contained one species, *Blaps lethifera*, with the highest mean *SCP* value. *SCP* was significantly different between groups (all comparisons *p* < 0.0001).

The average *SCP* of *Penthicus altaicus* beetles in autumn was significantly higher than that of those acclimated and exposed under −5 °C (*F*_1,59_ = 91.9, *p* < 0.001) (Table 4). The lowest values in these two samples differed by 1.5 times (−22.4 vs. −31.9 °C). In the autumn, 67% of individual *SCPs* were above average. In the sample of insects overwintered at −5 °C, on the contrary, only one value out of 43 was high (−11.6 °C), and 44% of individuals had an *SCP* above average. Thus, acclimation increased the cold resistance of *P. altaicus*.

The average *SCP* values of *Bioramix picipes* larvae were lower than those of the imago stage (−28.7 ± 0.3 and −21.7 ± 1.3 °C, respectively).

#### 3.1.2. Low Lethal Temperature

Not a single individual survived exposure for 2 days at a temperature below the minimum *SCP* value for the species (Table 5), indicating their inability to tolerate freezing.

Despite statistically indistinguishable *SCP* in a pair of species, *Anatolica dashidorzsi* and *Penthicus altaicus*, their resistance to long-term cooling was different; for example, the proportion of survivors after exposure to −20 °C differed by half—36 and 69%, respectively. In contrast, more than half of the sample of *Bioramix picipes* beetles survived a two-day exposure to −22 °C, although they had a slightly higher *SCP* (−21.7 °C) (Table 4 and Table 5).

*Blaps lethifera*’s resistance to prolonged exposure to negative temperatures was found to be low (only 10% of the sample survived at −15 °C), as might be expected because these beetles overwinter in mammal burrows.

### 3.2. Climatic Conditions in the Modern Range of Darkling Beetles

The three most cold-resistant species of darkling beetles are distributed in the most continental and cold part of Central Asia (44–52° N, 85–106° E), i.e., in the area of dominance of the Asian anticyclone (Figure 7).

In the Great Lakes Depression, the absolute minimum air temperatures drop to −48 °C, and in the depressions of Tuva and South-Eastern Altai, even the average of the absolute minimum temperatures have even lower values of −50 to −51 °C [30,32,34]. Snow depth is negligible; in January–February in the Mongolian part of the range and in the Chuya Depression of Altai, it barely reaches 5–10 cm [31,36]. This combination of air temperatures and snow depth leads to severe cooling of the soil.

Our mapping of long-term average minimum temperatures in the soil at a depth of 3 cm (Figure 8) showed, at first glance, an amazing uniformity of conditions in the elevational range from 500 to 2100 m a.s.l. over a vast area (up to the steppes of Transbaikalia): soil temperatures drop up to −18 to −22 °C. The reason for this phenomenon is an increase in air temperatures from north to south and a decrease of the snow depth in the same direction. In addition to the above parameters, the temperature of this mountainous area depends on the local topography, which controls temperature inversions and the accumulation of snow.

Outside the considered areas, winter soil temperatures are higher. For example, in the northwestern provinces of China, close to the eastern boundary of Gurbantonggut Desert in Xinjiang, experiences despite almost snow-free conditions [36], due to moderate air temperatures that rise to −13 or −14 °C. Soil temperatures are also higher in the mountain taiga of Central Altai due to the greater snow depth and higher of minimum air temperatures.

### 3.3. Overwintering Conditions for Darkling Beetles in Natural Habitats

#### 3.3.1. Temperatures in the Overwintering Microhabitat

The soil temperatures measured by loggers in the studied localities (Figure 9; Table 6) can be considered typical, since in the Kosh-Agach in the winters of 2021–2022 and 2022–2023, deviations from the long-term average values of minimum air temperatures and snow depth were small. Minimum soil temperatures at a depth of 10 cm in the dry and nearly snow-free area in the bottom of Chuya Depression near the Kosh-Agach Village dropped to −28 to −27 °C (logger 1-10). Soil temperatures were noticeably higher in the floodplain (−18 to −19 °C), where soil moisture and snow depth were greater (logger 2-10). On the southern slope of the Kuraisky Mt. Range, 510–520 m above the bottom of the depression, soil temperatures are primarily controlled by temperature inversions in the air. Due to the increase in minimum air temperatures with height (in winter 2021–2022 it was 9 °C), minimums in the soil at a depth of 10 cm did not fall below −17 to −18 °C (loggers 3-10, 4-10) (Table 6, Figure 9).

#### 3.3.2. Temperature Conditions in the Chuya Depression during the Last 90 Years

It should be borne in mind that the temperature situation described above is the result of significant climate change over the period of existence of the weather station. Annual minimum air temperatures in the Chuya Depression increased from 1934 to 2022 by 9–10 °C, and the depth of the snow, on the contrary, decreased by 3–4 cm (Figure 10). It is important that in cold winters the snowpack was above average, so that for 11 winters in which air temperatures dropped below −50 °C, the average depth of snow was 11 cm, and for 15 winters with temperatures above −40 °C it was only 3 cm.

Thus, the annual minimum of soil temperatures at a depth of 3 cm over the entire observation period increased by only 2–3 °C (Figure 10). The absolute minimum (−28 °C), reached in the winter of 1966–1967 was 5–7 °C lower than the minimum in the winters of 2021–2022 and 2022–2023 (−23 and −21 °C, respectively).

As mentioned above, two species of darkling beetles (*Blaps lethifera* and *Pedinus femoralis*) were collected near the Ongudai Village (weather station No. 13 in Figure 8). The climatic conditions here are different due to the greater influence of westerly winds in the cold season and lower absolute altitude (860 m). Minimum air temperatures are 9–10 °C higher, and the depth of snow in January is 9–10 cm (i.e., two-fold) deeper than in the Chuya Depression [30,31]. Therefore, the minimum soil temperatures at a depth of 3 cm are 6–7 °C higher than in the Kosh-Agach Village.

#### 3.3.3. Early Spring Observation of the Overwintering Microhabitats of Darkling Beetles

An early spring search (8 April 2023) for overwintered darkling beetles in South-Eastern Altai gave a positive result. Near the Kosh-Agach Village (the data logger 2), several hibernating imago of *Anatolica dashidorzsi* and *Penthicus altaicus* were dug out from the tussocks of *Achnatherum* and *Artemisia* from a depth of 7–10 cm. The beetles were then kept in plastic containers for more than two months, i.e., for them we can state a successful overwintering. Both species were also found at the northern border of the Chuya Depression at the foot of the Kuraisky Mt. Range. *Anatolica dashidorzsi* actively crawled along the southern slope during the daytime. Here, under small stones, awakened individuals of *P. altaicus* (the species is obligately nocturnal) were found, and a dormant specimen was dug out from an Astragalus turf from a depth of 10 cm.

Logger 4, installed near Tabozhok Mt. in the habitat of *Bioramix picipes*, was still under a 20–30 cm layer of snow at that time. Only a single specimen of this species was found at the edge of a snow field under a small stone, and this beetle also overwintered successfully.

## 4. Discussion

### 4.1. Cold-Resistance Parameters of Darkling Beetles

#### 4.1.1. Supercooling Point

It is known that darkling beetles use a variety of cold resistance strategies [14,38]. The results of our work allow us to classify the studied species as freeze-intolerant poikilothermic species that use supercooling mechanisms to protect themselves from negative temperatures. Some expected properties of the studied darkling beetles were also observed. Like most insects from regions with seasonal temperature changes [39,40,41,42], *Penthicus altaicus* shows an increase in cold resistance (judging by *SCP*) from autumn to winter (from −13.1 to −25.0 °C). Unfortunately, the named species is the only one that was collected in sufficient numbers to determine cold resistance in two seasons.

The resistance to cold in both larvae and adults of *Bioramix picipes* was also expected, since both stages overwinter in the soil. Greater cold resistance of larvae compared to imago (−28.7 and −21.7 °C, respectively) is characteristic of many insects; numerous cases of thermotolerance change during ontogenetic development are widely described [43,44].

The range of average *SCP* values of the darkling beetles we studied (*Anatolica dashidorzsi*, *Penthicus altaicus*, and *Bioramix picipes*) is only 4 °C: from −26 to −22 °C (Table 4). It is similar to the *SCP* of the darkling beetle *Microdera punctipennis* Kaszab, 1967. This species is known from the Gurbantonggut Desert in Xinjiang and has an average *SCP* in December of −18.7 °C (the lowest individual value is about −20 °C). *Microdera punctipennis* overwinters at −15 °C (measured at collection site in December) at a depth of more than 5 cm in sand under shrub leaf litter [13,45]. Note that according to long-term data, winter in the Gurbantonggut Desert is warmer than in Northern Mongolia (Figure 7).

#### 4.1.2. Comparison of SCP and LLT

It is known that the *SCP* value does not always reflect the true resistance to negative temperatures. Its value in animals overwintering in a supercooled state is, as a rule, below the limits of tolerable temperatures [14,46]. At the same time, *SCP* is an easily measured and therefore statistically reliable value. It is difficult to obtain the same survival characteristics of a large samples of animals at different temperatures. This raises problems of inconsistency between cold tolerance parameters and interpretations.

In our experiments, for example, despite the equality of *SCP* (−25 °C) in *Anatolica dashidorzsi* and *Penthicus altaicus*, the survival of these beetles after long-term exposure at −20 °C differs almost two-fold at 36 and 69%, respectively (Table 4 and Table 5). A possible explanation follows from the analysis of the results of repeated determinations of beetle survival in the control (at −5 °C). Two replicate controls of *Bioramix picipes* differed from each other by only 5%, while controls of *A. dashidorzsi* differed by 10%, and controls of *P. altaicus* differed by 26% (Table 5). The samples were, of course, randomly selected and the containers were kept in the same test chamber. However, the samples were collected with difficulty and did not allow the use of a large number of replicates.

Heterogeneity in samples can arise for many reasons and can be due, for example, to a different proportion in the population (i.e., in collected materials) of a given species of old beetles that have survived until their final autumn. It also cannot be ruled out that populations of species are not equally conditioned for wintering (in the current year), etc. However, the absence of errors in the formation of samples and the results obtained is suggested by the normality of the distributions of *SCP* adults of all darkling beetle species.

The extensive materials used in this work on *SCP* (177 measurements) and *LLT* (20 samples of four species, a total of 456 individuals) do not contain obvious artifacts. We are aware that the results for cold resistance are not sufficient to identify the features of the physiological mechanisms of insects that have been so little studied. Available data make it possible to estimate a common range of low lethal temperatures for three species of darkling beetles (i.e., except *Blaps lethifera* and *Pedinus femoralis*) of −22 to −20 °C, at which 50% of individuals die in the experiment after exposure for up to two days (Table 5). Based on the ratio of this parameter and the temperatures of *SCP* (average and minimum), the 50% threshold of *LLT* of −20 °C for *Anatolica dashidorzsi* and *Penthicus altaicus* seems to have been overestimated, and a value closer to reality is around −25 °C.

### 4.2. Climate in the Region

#### 4.2.1. Bottom of the Chuya Depression

First, we note that the Chuya Depression of the Altai Mountains is a convenient testing ground for studying the cold resistance of darkling beetles. At the bottom of the depression, soil temperatures turned out to be close to the lowest in the arid part of Central Asia at altitudes less than 2000 m above sea level [47].

The climate of the Chuya Depression is harsh, with average annual air temperatures close to −5 °C [29], and together with an insignificant snowpack, they form permafrost. this completely underlies the depression, and small areas without it are confined to river floodplains, lake bottoms, etc. Due to climate warming, permafrost is degrading [29,30], and the seasonally thawed layer now reaches 3.7 m, in some places even 6.5 m [48,49]. The soil already thaws up to 40 cm in early April, and up to 160 cm by mid-May [29,30]. Therefore, permafrost layers do not affect the temperature of the upper layers of the soil and may not be taken into account in this work. The described permafrost situation is very close to that known for the relict steppes of Northeast Asia [6].

The Chuya Depression, as a cold testing ground, turns out to be representative for a significant part of Central Asia. As we noted, to the south of these territories, air temperatures rise in winter and the amount of precipitation decreases, which is reflected in a slight variation in soil temperatures over a vast area. On the contrary, to the north, in a broad sense, in Siberia, minimum air temperatures remain at the level of −45 to −40 °C, but early falling snow and deep snowpack protect the soil from significant cooling. Background values for minimum soil temperatures at a depth of 20 cm, for example, north of Tomsk, are only −5 to −7 °C [50].

#### 4.2.2. Mountain Habitat around the Chuya Depression

The lack of weather stations in the mountain ranges surrounding the Chuya Depression does not allow the analysis of extreme temperature values over long-term observations. Therefore, to assess conditions, it is necessary to extrapolate data from a few data loggers. In this work, we examined only the southern slope of the Kuraisky Mt. Range, which is preferred by darkling beetles [20]. The location of data loggers in the most contrasting conditions, which are warmed xeromorphic versus shaded and relatively humid microhabitats, allows us to estimate the variation of temperature values.

The climate of the mountains surrounding the Chuya Depression is much milder. There is a pronounced temperature inversion here, and at higher altitudes winter temperatures are noticeably higher than at the bottom. This effect is clearly visible on the loggers we installed (Table 6). When rising 500 m above the bottom, minimum temperatures increase by 8–10 °C. It is very likely that inversions may be responsible for the greater species diversity of darkling beetles on mountain slopes in comparison with the flat part of the depression.

However, winter soil temperatures depend not only on air temperature, but also on the snow depth and slope exposure. Moreover, in the mountains the snow is distributed even more unevenly than in the depression, and its average depth is much greater [51]. First, in general, precipitation increases with elevation. Secondly, snow begins to fall in the mountains somewhat earlier than at low elevations, often in September. Moreover, in the region, the monthly precipitation rate in the autumn months is two to three times greater than in the winter [18]. Thirdly, thanks to the topography, there are significantly more “traps” for snow when it is transported horizontally.

The 2021–2022 season, according to the Kosh-Agach weather station, was characterized by a small amount of snow, and the amount of precipitation from October to February was three times less than normal, and in November, January and February there was no precipitation at all [18]. But even in this season, loggers 4-1 and 4-10 installed at the foot of the northern slope, judging by the small daily range of temperature, were under snow from early October to mid-May (Figure 9). At a depth of 10 cm, there was a gradual (without sudden changes) decrease in temperature; in mid-February a minimum was reached (about −18 °C), after which there was a gradual rise in temperature (Figure 9, Table 6). Obviously, in winters with more snow, soil temperatures should be even higher in these habitats, and their changes even slower.

Less predictable are the winter readings of loggers 3-1 and 3-10, installed on a south-facing-slope. On the one hand, this slope receives a large amount of solar radiation, on the other hand, in the absence of snow, it is subject to rapid cooling. In the low-snow season of 2021–2022, judging by the daily range of temperature, the studied section of the slope was covered with snow for about a month only, from mid-December to mid-January, and from about January 20 it was snowless. As can be seen from the graph (Figure 9), fluctuations in average daily temperatures on this slope during winter were quite significant compared to logger 4. It is noteworthy that at the beginning of winter, average daily temperatures at a depth of 10 cm in “snowless” conditions, and locations with snowpack, were close to each other and were −5 to −10 °C in November and −10 to −12 °C in December. At the end of January, the southern-facing slope began to warm up, and in February, logger 3-10 showed higher temperatures (both the average and the minimum) than logger 4-10, which was covered by snow.

In contrast, in the 2022–2023 season, large amounts of precipitation fell during the winter. In Kosh-Agach weather station, 12 mm (356% of the norm) was measured in December, and 5 mm (161% of the norm) in January [18]. This turned out to be sufficient for the microhabitat where logger 3 was installed to remain under snow all winter. The pattern of change in temperature of the upper soil layers in the winter of 2022–2023 was similar to that described for logger 4 in the previous year. The minimum temperature at a depth of 10 cm dropped in January to −17.5 °C, i.e., to approximately the same values as in the low snow season.

The results described must be extrapolated with great caution. Solar heating of soils in different areas depends on many factors, such as exposure, surface albedo, precipitation and wind regimes, and orography, etc. However, the value of the information obtained lies in the fact that it shows a noticeable temperature effect from the influence of solar radiation on the snow-free soil surface, comparable to the influence of snowpack. The radiation regime of South-Eastern Altai, as a high-mountain region, is favorable for the formation of such an effect. The global solar radiation at the Kosh-Agach weather station, even in December, averages 133 MJ/m^2^, and by February it increases to 274 MJ/m^2^, which is 2 and 1.5 times more than in Volgograd City, located at the same latitude, respectively [17,52].

### 4.3. Cold Resistance of Darkling Beetles and Overwintering Temperatures

#### 4.3.1. Contradiction between the Cold Resistance of Darkling Beetles and Regional Conditions

An important result from the previous section is that in the Chuya Depression, as well as in the vast expanse of the north of Central Asia, the average long-term minimum temperatures at a depth of 3 cm in the soil lie in a relatively narrow range; its lower limit is −24 °C, and in very cold years the minimums can drop to −30 to −28 °C (Figure 10). It is clear that the tolerance of darkling beetles to negative temperatures is insufficient for persistence at a depth of 3 cm, and that, at a depth of 10 cm, successful overwintering is not possible in some microhabitats.

The usual situation for widespread and eurytopic species living in extreme regions is exactly the opposite. They have a significant “reserve of cold resistance”, which is the difference between the low limit of beetle resistance and minimum temperatures in the overwintering places [42]. The results obtained for the darkling beetles from the Chuya Depression (as well as *Microdera punctipennis* from the Gurbantonggut Desert) do not provide evidence for such a reserve. But the existence of stable populations without a “safety cushion” seem unlikely, because the very first climate fluctuation towards cooling would lead to collapse. Therefore, this raises a question about how to interpret measurements of cold resistance.

We do not see any possible fundamental (instrumental or protocol) errors in the method for determining cold resistance, which is routinely used and has been tested on dozens of species of invertebrate animals [26,42,53,54,55]. At the same time, there are surprisingly few publications on the cold resistance of desert and steppe darkling beetles (with the exception of crop pest species), which indicates limited experience in working with them in nature and in the laboratory [13,14,15,16].

In our case, nothing is known about the potential for the timing of collection of these beetles to determine cold resistance. We collected them on August 16–18 (Figure 11). Starting from this moment, minimum air temperatures fell into the range of 0–5 °C, and from the beginning of the second ten days of September they became mostly negative or close to this. However, average daily temperatures remained high for more than a month. We believe that the above figure indicates the correct choice of time for collecting insects. Moreover, it is much more difficult to collect sufficient material at a later date due to a decrease in beetle activity. At the same time, it is possible that the significant heterogeneity in the quality of the samples, revealed in the control of experiments when assessing the low lethal temperatures, is associated with the early period of beetle collection. For *Penthicus altaicus*, survival varies in the range of 26% (55–81%), which cannot be ignored when assessing survival parameters.

It is also possible that there are factors that we do not take into account when acclimating insects in the laboratory that could influence the identified cold resistance.

But the main problem probably lies in the literal overlap of available data. By default, it is assumed that beetles overwinter exactly where the loggers are installed, up to a depth of 10 cm; temperatures in deeper soil horizons were estimated using long-term average data from the Kosh-Agach weather station. With this method of assessing temperature conditions, successful overwintering of darkling beetles in the immediate surroundings of the weather station in the period from 1989 to 2022 would have been impossible: the average minimum soil temperatures to a depth of 50 cm were below the critical temperature for darkling beetles (−22 °C) [29].

#### 4.3.2. Heterogeneity of the Topography and Vegetation within the Territory

Contrary to the described temperature conditions, darkling beetles of the studied species are very numerous in the Chuya Depression. The main contradiction, in our opinion, lies in the underestimation of the heterogeneity of the temperature distribution associated with microrelief, soils, and the presence of rivers, etc.

Microrelief in winter primarily affects the distribution of the scarce snow, as well as soil temperatures. The weather station, located on a terrace above the floodplain, is adjacent to a vast, slightly sloping plain (a plume of mountain slopes), replete with flat depressions of varying widths and depths. There is more snow in the hollows, both due to the wind shadow and greater projective cover of vegetation (here in the summer the soil moisture is slightly higher), which retains the snow. In the hollows, in a year with normal precipitation, the snow depth is about 15 cm (in a snowy year it is around 40 cm) [29], which is twice as much as at the weather station (Table 7). Under such conditions, at a depth of 3 cm, the minimum soil temperature will be at least 4 °C higher [28]. Due to the fact that, in the Kosh-Agach, the minimum temperatures at depths from 20 to 40 cm increase by 5 °C, the total temperature increase in the hollows will be about 9 °C.

Favorite shelters for darkling beetles of many species in the region are the spaces under stones, as well as within plant litter that accumulates at the base of plant stems (especially *Caragana* and *Achnatherum*) [20]. Such shelters, as well as constituting microrelief, serve as snow traps, and plant litter can be considered as an additional insulating layer. For example, we recorded the successful overwintering of *A. dashidorzsi* and *P. altaicus* at a shallow depth (7–10 cm) in the *Achnatherum* sods. Snow can also accumulate around medium and large stones, and the stones themselves, in the absence of snow, act as a temperature buffer. While rivers and lakes are a known source of heat for permafrost areas in winter [56], their role in the overwintering of xerophylic darkling beetles is not obvious.

Four species of the genus *Blaps* are known from the South-Eastern Altai [12], and all use mammalian burrows for overwintering microhabitats [11]. Their more modest cold resistance is to be expected, given the warmer conditions in these burrows.

The conclusion that we have reached is that adults and larvae of darkling beetles in the Chuya Depression can successfully overwinter only in warm microhabitats. The most likely overwintering place for *A. dashidorzsi* and *P. altaicus* is in the spaces under bushes with abundant loose litter and in the tillering nodes of large cereals, especially those located in shallow flat and gentle hollows. For other species, microhabitats can apparently be fundamentally different, as for their survival strategies.

#### 4.3.3. Overwintering Depth

A separate subject of discussion remains the question of the overwintering depth of beetles and larvae of Tenebrionidae species living on the bottom of the Chuya Depression, in particular: is the large overwintering depth a main factor providing a “cold resistance reserve?”.

No specific research on this topic has been carried out in the Chuya Depression. The ability to burrow in light (sandy) soils is undoubtedly present even in non-specialized species (*Penthicus altaicus*, *Anatolica dashidorzsi*, and *A. strigosa* (Germar, 1823)). At the same time, a number of arguments can be made that challenge the possibility of their overwintering in deep layers. Firstly, excavations of darkling beetles of various species, including in the adjacent Uvs Nuur Depression, showed that even species adapted to digging burrow into the soil to a depth of no more than 15 cm [11]. Both adults and larvae of well-studied steppe pest species from Eastern Europe also inhabit surface soil layers [19]. This is consistent with our small experience of collecting darkling beetles from Altai, including in the Chuya steppe. Secondly, most of the species of darkling beetles in the South-Eastern Altai and adjacent Tuva and Mongolia are confined to gravelly microhabitats, with only small patches of sand at the base of plant bushes [20]; such species include *P. altaicus*. Usually these species hide under shelters and do not bury themselves at all. Species of the genus *Anatolica* characteristic of the Chuya Depression are also poorly adapted to digging [11]. The early spring activity of *P. altaicus* and *A. dashidorzsi* that we observed is essentially proof of the possibility of their overwintering in the upper layers of the soil.

#### 4.3.4. Recovery of Darkling Beetle Populations after Extremely Cold Winters

The options proposed above for explaining the consequences of the phenomenon of the lack of a “reserve of cold resistance” in the darkling beetles of the Chuya Depression can be acceptable for minor fluctuations in the minimum temperatures of different years. However, abnormally cold winters with minimum air temperatures below −50 °C [18,51], and soils at a depth of 3 cm cooling down to −28 °C are known even during meteorological observations (the last 90 years). It is very likely that in historical times (for example, during the Little Ice Age) winters here were even colder. After such winters, we can expect a reduction in the number of darkling beetles, probably up to the complete disappearance of some species.

The duration of depletion in numbers may depend on the depth of “temperature damage” of the population and the peculiarities of the species biology. The abundance may recover due to larvae hibernating in the soil stratum. The second source of population recovery at the bottom of the trough may be darkling beetles dispersing from lake depressions and from mountain slopes with southern exposure. A significantly greater species diversity of darkling beetles was noted on the slopes compared to the depressions of both South-Eastern Altai and the adjacent regions of Tuva and Mongolia [11,20]. On the contrary, all steppe species of darkling beetles from the bottom of the Chuya Depression are also known to be found in the surrounding mountains [12]. On mountain slopes with high habitat heterogeneity, milder conditions develop in winter, primarily due to temperature inversions as well as topography and exposure. The areas affected by temperature inversions are large. In the mountainous regions of Central Asia, the thickness of the layer with a temperature inversion reaches 1000 m above the bottom of the depressions, while air temperatures in the coldest periods rise by 1 °C per 100–120 m elevation increase [57]. Comfortable temperatures in areas with thicker snowpacks are suitable for mesoxerophilic species such as *Bioramix picipes*. As shown above, higher average daily soil temperatures were also observed on heated slopes with southern exposure, suitable for the habitat of most species in the region.

Thus, there are two sources of restoration of darkling beetle populations on the bottom of depressions after extremely cold winters: beetle larvae overwintering in the soil, and adults dispersing from lake vicinities and from the mountain surroundings. Unfortunately, the limited information available on interannual fluctuations in the number of darkling beetles does not allow direct confirmation of these considerations.

### 4.4. Climatic Conditions of the Late Pleistocene of Siberia and the Darkling Beetle

Insects of late Pleistocene deposits in Siberia have been studied extremely unevenly. Numerous entomological assemblages (with a predominance of Coleoptera) have been described from the territory of the West Siberian Plain and Northeast Siberia; only singleton records are known from other regions [5]. It is expected that darkling beetles are absent in the late Pleistocene deposits of the northern part of Western Siberia, where reconstructed conditions are similar to those of the modern Arctic [58]. The absence of this family in the Pleistocene of Northeast Siberia and their very weak representation in the southern part of the West Siberian Plain require discussion.

#### 4.4.1. South of the West Siberian Plain

Open periglacial tundra-steppe, forest-steppe, or steppe landscapes existed in the south of the West Siberian Plain (south of 60° N) at the end of the late Pleistocene. They were formed in a cold and dry climate [59]. There are no analogous landscapes in the modern zonation system, so obtaining quantitative climate assessments for these communities is extremely difficult. This is especially true for winter temperatures, the reliability of reconstruction of which is low even for modern landscapes, and a weak correlation of reconstructed and actual temperatures is shown by both the analogue method [60] and the Mutual Climatic Range method [61,62].

Reliable reconstructions were obtained for the southern part of the region—the basin of Lake Aksor (53° N). The climate here was characterized by instability, as eight cycles of alternating cryogenic extremely dry and cold epochs with moderately cold conditions have been identified in the section during marine isotope stage 2 (MIS 2). The average annual temperature of the cryogenic times is estimated to have been 13–21° colder than in modern times (i.e., from −9 to −17 °C), with average annual precipitation less than 100 mm, and winters characterized by little snow and strong winds [63,64].

Late Pleistocene assemblages of Coleoptera in this region have been characterized in 16 localities ranging in age from the second half of MIS 3 to the end of MIS 2 (Last Glacial Maximum), located at latitudes from 51.5 to 58° N [9,58,65,66]. The identified fauna, including more than 400 species of Coleoptera, has no modern analogue on the plain, and reconstructions made on its basis are in good agreement with paleobotanical and soil reconstructions of periglacial steppes. The closest modern analogues are the entomofauna of the intermountain steppe depressions of Southern Siberia and Northern Mongolia, especially South-Eastern Altai [7,8,9]. The South-Eastern Altai is also cited as the closest modern analogue of the periglacial tundra-steppes of Eurasia based on an analysis of vegetation, mammal faunas, and terrestrial molluscs [67].

At the same time, two important differences do not allow us to fully “transfer” the modern natural environment of South-Eastern Altai to the late Pleistocene of the West Siberian Plain. Firstly, some species extracted from one taphocoenosis at present inhabit different altitudinal zones of the South-Eastern Altai and do not occur together. Their cohabitation in the Pleistocene of the West Siberian Plain indicates a greater continental climate (for more details, see ref. [9]). Taking into account the reconstruction of average annual temperatures 5–13° colder than now in Kosh-Agach Village and greater continentality, the difference in winter temperatures should be even more significant.

The second difference is the representation of the family Tenebrionidae. Darkling beetles from the subfamilies Pimeliinae and Tenebrioninae are represented by 15 species in South-Eastern Altai, and in terms of abundance they are one of the dominant groups in all arid landscapes [12]. In the periglacial steppes of the late Pleistocene of the region, only four species from these subfamilies are known, and they are represented by singletons: locality Suzun-1 (age ~25.5 cal kyr BP) in the Upper Ob region—*Scythis* sp. and *Platyscelis* sp.; and locality Gornovo-IV (MIS 3) from the Southern Cis-Urals—*Anatolica abbreviata* (Gebler, 1832) and *Pedinus* cf. *femoralis* [9,66]. Note that the Coleoptera assemblages of these localities, although quite typical for the fauna of periglacial steppes, are still distinguished by high species diversity (each has more than 110 species) and the diversity of reconstructed landscapes. Taking all of this into account, it can be assumed that extreme low temperatures in winter are one of the factors limiting the distribution of darkling beetles during the cold periods of the Pleistocene. Probably, during the coldest epochs of the last glaciation, darkling beetles could not exist in the south of the West Siberian Plain. During the relatively mild epochs of this period, as well as the MIS 3 interglacial period, some species of darkling beetles could overwinter only in relatively warm habitats, and therefore are present in sediments in small numbers.

As an exception, halobiont darkling beetles of the genus *Centorus* from the subfamily Lagriinae are characteristic of the periglacial fauna of the region (known from six localities, abundant in some) [8,9,66]. However, their cold resistance and habitat conditions have not yet been studied.

#### 4.4.2. Northeast Siberia

Even more severe conditions developed in the basins of the Indigirka and Kolyma rivers (i.e., to the north at 13–15° N and significantly to the east) in xeromorphic communities with steppe insect species. Here, the lowest air temperatures in winter drop to as low as −63 or −61 °C, which is about 10 °C lower than in the Kosh-Agach Village or in the intermontane depressions of Tuva. Minimum temperatures in the soils of xeromorphic communities in the upper Kolyma, whose summer temperature conditions are similar to those of the Chuya Depression, vary widely depending on the snow thickness. In the least snowy areas, snow depth does not exceed 20–25 cm, which is almost twice as much as in the depressions of South Siberia and Mongolia, such as the Chuya and the Great Lakes regions. In the first centimeter of soil, the lowest temperature can drop to −43 °C (with a monthly average of about −26 °C), and at a depth of 20 cm, the same pair of indicators is −33 and −21 °C.

Similar soil temperatures in South-Eastern Altai and in xeromorphic communities of Northeast Asia indicate the theoretical possibility of the existence of some species of darkling beetles in the north-east. However, they are not here, as we believe, due to paleogeographical reasons.

The reconstructed average air temperature of the coldest month during the Last Glacial Maximum in the Kolyma basin was −48 °C [68,69,70], which allows us to estimate minimums of −60 to −65 °C. The thickness of the snowpack was less than modern, so lowest soil temperatures should have dropped below −40 °C.

Based on the above estimates, it can be assumed that the studied species could not have existed in the arid late Pleistocene landscapes of Northeast Asia.

## 5. Conclusions

A study of the cold resistance of five species of imago and one species of larvae of Altai darkling beetles showed that they all use the supercooling mechanism to protect themselves from negative temperatures and do not endure freezing. The average *SCP* value for adults of *Anatolica dashidorzsi*, *Penthicus altaicus*, and *Bioramix picipes*, which are the most cold-resistant of the studied species, range from −25.7 to −21.7 °C. Their most cold-resistant individuals are supercooled to temperatures between −35 and −30 °C. However, the lowest temperatures tolerated by *A. dashidorzsi* and *P. altaicus* for up to two days are noticeably higher, as 50% of individuals die after exposure to temperatures of about −20 °C. Based on a comparison of this indicator and *SCP* (average and minimum), a 50% threshold of low lethal temperatures of −20 °C seems to be too high.

The ranges of the three mentioned species of darkling beetles are limited to the area in the northern desert-steppe part of Central Asia with an extreme continental climate. The Chuya Depression of South-Eastern Altai is one of the coldest places of the territory, with extremely low winter soil temperatures having been reconstructed from weather station data. Here, the minimum soil temperatures at a depth of 10 cm in natural beetle habitats are somewhat lower than the lowest lethal temperature of darkling beetles, even in winters with average temperature conditions (2021–2022 and 2022–2023).

The absence of a “cold resistance reserve”, the difference between wintering temperature conditions and cold resistance, makes the existence of darkling beetles of the studied species contradictory. This contradiction is less pronounced in the mountain setting, where conditions, in general, are much milder than in the depressions due to the inversion of winter air temperatures. In addition, the mountains are characterized by the uneven distribution of snow and by solar heating of snowless slopes with southern exposure. Minimum soil temperatures at a depth of 10 cm, determined by data loggers at an altitude of ~500 m above the bottom of the Chuya Depression, are quite suitable for the successful overwintering of darkling beetles.

A few species of darkling beetles (of those studied are *Anatolica dashidorzsi* and *Penthicus altaicus*) are known from the bottom of the Chuya Depression, the coldest place in the region. They are obviously confined to microhabitats that are warmer in winter. The most likely overwintering place is the space under shrubs (mainly *Caragana* spp.) with abundant loose litter and tillering nodes of large cereals (*Achnatherum* spp., etc.), especially those located in shallow flat and gentle hollows. Such plants retain snow, which together raises temperatures to a level sufficient for the successful overwintering of darkling beetles. There is a need to improve our information about the places and depth of overwintering of imago and larvae of darkling beetles of different species to confirm our intepretations.

Thus, the work carried out shows that darkling beetles are constrained by physiological limits to inhabit specific microhabitats in the Chuya Depression. We emphasize that the studied species find specific microhabitats in order to persist in the most severe conditions possible in Central Asia at altitudes up to 2000 m a.s.l. Significantly lower temperatures during the cold stages of the late Pleistocene both in the Chuya Depression and on the West Siberian Plain, and even more so in Northeast Asia, were unsuitable for darkling beetles. In this regard, it is natural that darkling beetles are now absent in the relict steppes of the Northeast, a derivative of the Pleistocene tundra-steppes.

## Figures and Tables

**Figure 1 insects-15-00064-f001:**
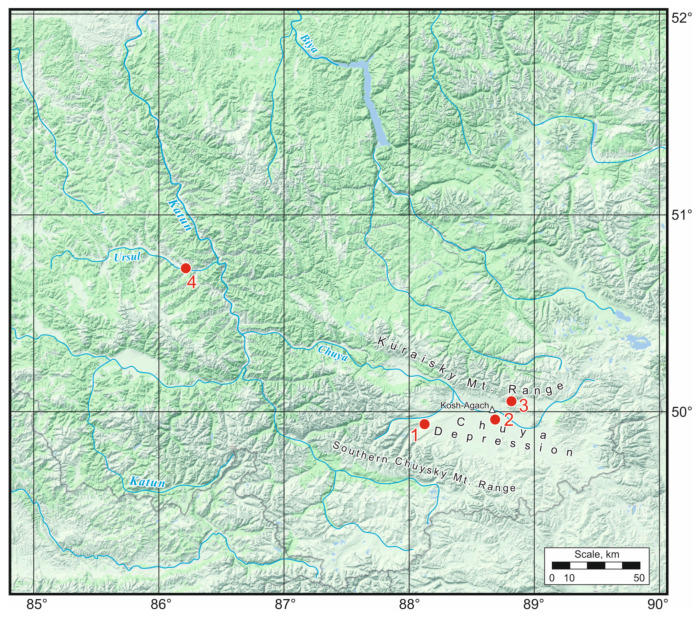
Collection sites of the studied species of darkling beetle. 1—Beltir Village, 2—Kosh-Agach Village, 3—Tabozhok Mt., and 4—Ongudai Village.

**Figure 2 insects-15-00064-f002:**
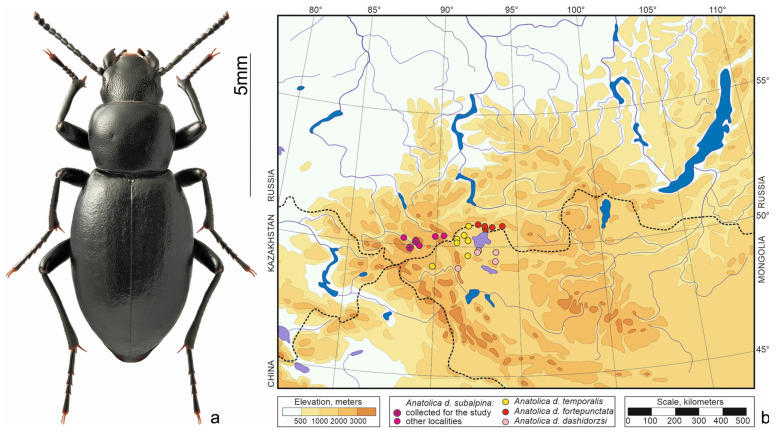
*Anatolica dashidorzsi*. (**a**) Habitus of the beetle from Kosh-Agach, photo by S.V. Reshetnikov; (**b**) distribution map according to [12,20] and collections of Siberian Zoologica Muzeum, Institute of Systematics and Ecology of Animals (SZMN).

**Figure 3 insects-15-00064-f003:**
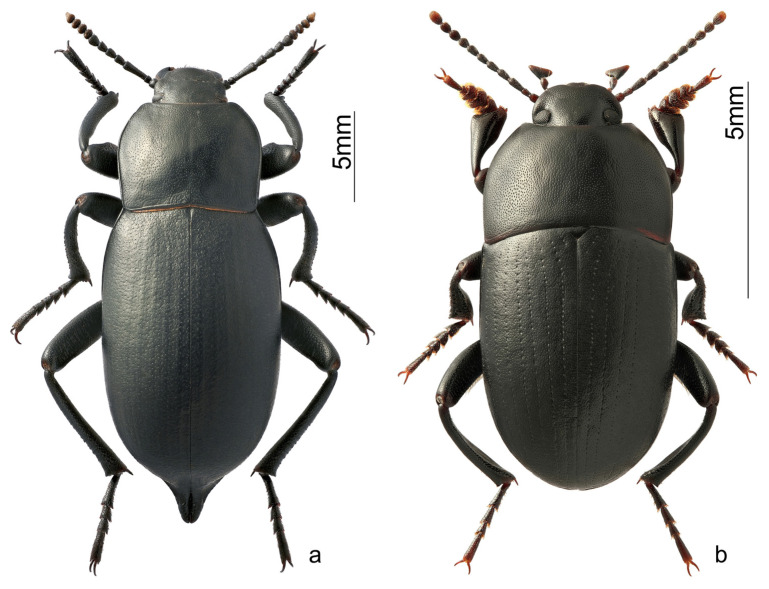
Habiti of the darkling beetles from Ongudai, photo by S.V. Reshetnikov; (**a**) *Blaps lethifera*; and (**b**) *Pedinus femoralis*.

**Figure 4 insects-15-00064-f004:**
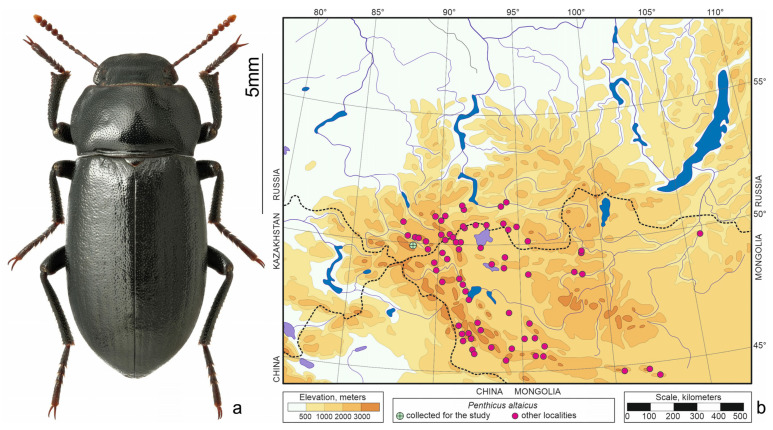
*Penthicus altaicus*. (**a**) Habitus of the beetle from Kosh-Agach, photo by S.V. Reshetnikov; (**b**) distribution map according to [20,23,24] and collections of SZMN.

**Figure 5 insects-15-00064-f005:**
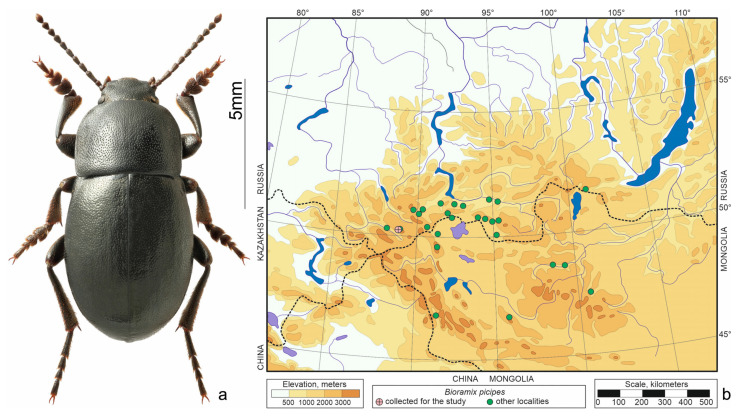
*Bioramix picipes*. (**a**) Habitus of the beetle from Tabozhok, photo by S.V. Reshetnikov; (**b**) distribution map according to [20,25] and collections of SZMN.

**Figure 6 insects-15-00064-f006:**
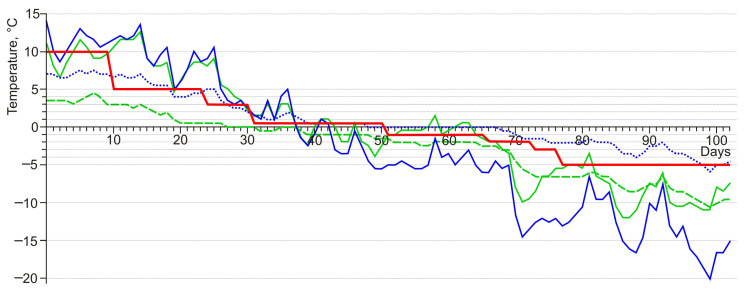
Graphs of darkling beetle acclimation in the experiment (red line) and lowest daily soil temperatures at a depth of 10 cm in the beetle habitats. Green lines—the foot of Tabozhok Mount; blue lines—surroundings of the Kosh-Agach Village; solid lines—positive relief elements with xeromorphic vegetation; dotted line—river floodplain; dashed line—foot of the slope with deep snow.

**Figure 7 insects-15-00064-f007:**
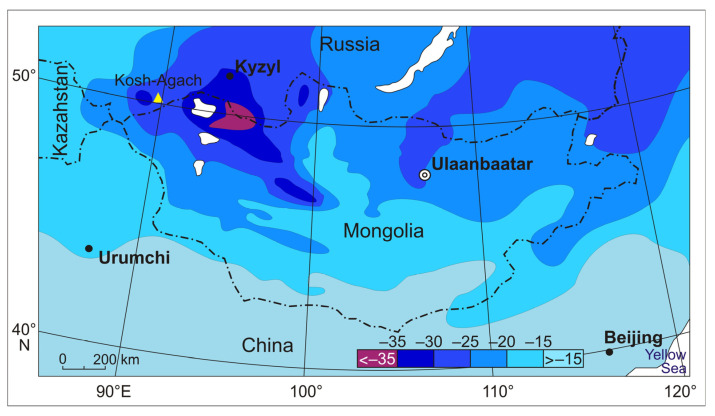
Average January air temperatures in northern Central Asia [35]. The studied area is near the yellow triangle (Kosh-Agach Village).

**Figure 8 insects-15-00064-f008:**
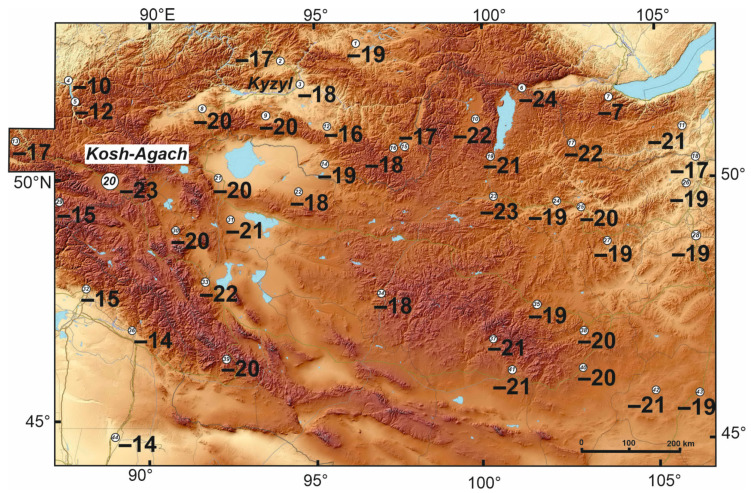
Long-term average of minimum temperatures (°C) at a depth of 3 cm, calculated from weather station data. The big white circle is a place for the darkling beetle collection and for installation of the data loggers. The weather station numbers are circled and, their names and altitudse (m) are as follows: 1—Toro-Hem (920), 2—Turan (862), 3—Kyzyl (628), 4—Yailu (482), 5—Belya (560), 6—Mondi (1304), 7—Hamar-Daban (1442), 8—Chadan (832), 9—Ak-Tal (1030), 10—Rinchinlhumbe (1583), 11—Iro (656), 12—Znamenka (606), 13—Ongudai (860), 14—Erzin (1101), 15—Kungurtuk (1310), 16—Chirgalandi (1386), 17—Sanaga (1170), 18—Kyahta (791), 19—Khatgal (1624), 20—Kosh-Agach (1759), 21—Ulaangom (936), 22—Baruunturuun (1232), 23—Murun (1283), 24—Tarialan (1235), 25—Hutag-Ondor (940), 26—Bayangol (800), 27—Bulgan (1250), 28—Baruunhara (800), 29—Katanda (950), 30—Ulgii (1715), 31—Omnogovi (1590), 32—Altai (737), 33—Hovd (1405), 34—Uliastai (840), 35—Tsetserlag (1684), 36—FuYun (827), 37—Galuut (2110), 38—Hujirt (1760), 39—Baitag (1186), 40—Arvaikheer (1813), 41—Bayankhongor (1500), 42—Sayhan-Ovoo (1316), 43—Mandalgovi (1350), and 44—Qitai (790).

**Figure 9 insects-15-00064-f009:**
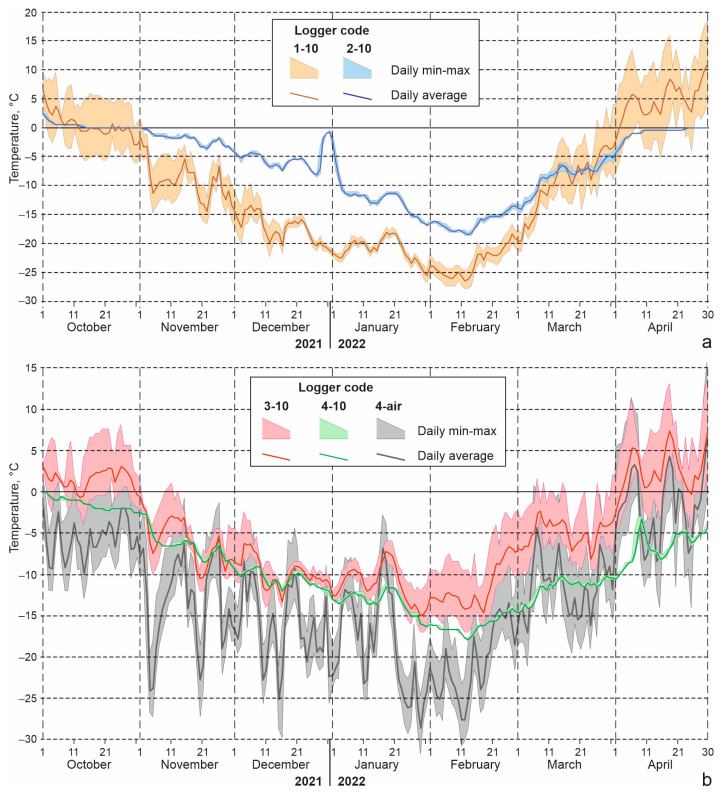
Minimum, maximum and average daily temperatures of air and soil at a depth of 10 cm. Loggers were installed in the Chuya Depression (**a**) and in the southern macroslope of the Kuraisky Mt. Range (**b**); for the locations of the loggers, see Table 3.

**Figure 10 insects-15-00064-f010:**
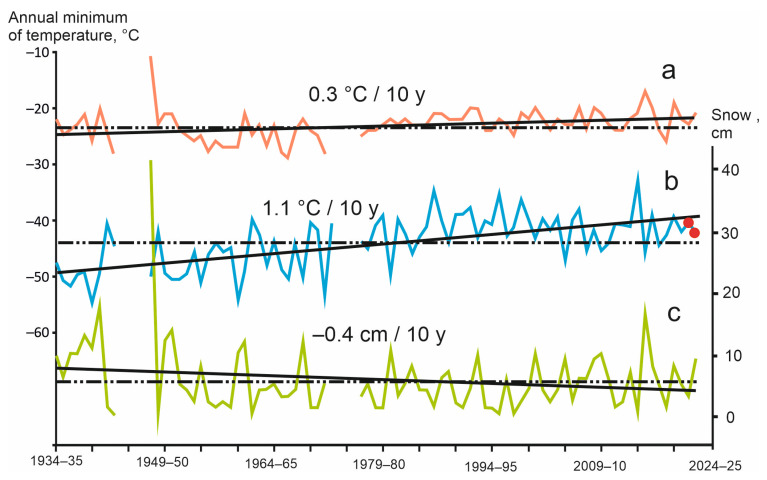
Long-term variation of climatic parameters: annual minimum of soil (**a**) and air (**b**) temperature; and (**c**) snow depths [29,37]. Red circles are years of soil temperature measurements by the data loggers, solid lines are long-term trends of the parameter, dash-dotted lines are average values of the parameter for the period of operation of the weather station.

**Figure 11 insects-15-00064-f011:**
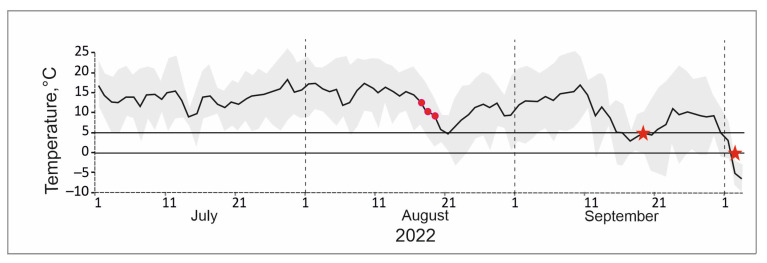
Dynamics of average (solid line) and extreme (shading) daily temperatures in July–September 2022. Circles indicate the dates of collection of darkling beetles, and asterisks indicate the long-term average dates of transition of the mean daily temperature through 5 and 0 °C [30].

**Table 1 insects-15-00064-t001:** Collection sites for 5 species of darkling beetles.

Species, Stage *	Collection Site	Height A.S.L., m	Number of Specimens
*Anatolica dashidorzsi*, Im	Beltir Village, 49.931° N, 88.117° E	1970	123
	Kosh-Agach Village, 49.941° N, 88.706° E	1760	
*Blaps lethifera*, Im	Ongudai Village, 50.725° N, 86.206° E	800	120
*Pedinus femoralis*, Im			31
*Penthicus altaicus*, Im	Beltir Village, 49.931° N, 88.117° E	1970	219
*Bioramix picipes*, Im	Tabozhok Mt., 50.061° N, 88.807° E	2250	133
*Bioramix picipes*, L			7

* Stages: Im—Imago, L—larvae.

**Table 2 insects-15-00064-t002:** Parameters of cold resistance studied in five species of darkling beetles.

Species, Stage	*SCP* *	*LLT* *
*Anatolica dashidorzsi*, Im	+	+
*Penthicus altaicus*, Im	+	+
*Pedinus femoralis*, Im	+	−
*Blaps lethifera*, Im	+	+
*Bioramix picipes*, Im	+	+
*Bioramix picipes*, L	+	−

* the parameter was determined (+) or not determined (−).

**Table 3 insects-15-00064-t003:** Locations of data loggers to determine soil and air temperatures in the study area.

Locality	Logger No.	Depth/Height Above Ground Level, cm	Season	Coordinates	Elevation A.S.L., m	Relief, Soil
Chuya Depression, Kosh-Agach Village	1-1	−1	2021–2022; 2022–2023	49.9422° N, 88.7069° E	1761	Positive relief element, xeromorphic
	1-10	−10				
	2-air	150	2021–2022	49.9474° N, 88.7087° E	1759	Floodplain of the Chaganka River, wet
	2-1	−1				
	2-10	−10				
Kuraisky Mt. Range, Tabozhok Mt.	3-1	−1	2021–2022; 2022–2023	50.0615° N, 88.8101° E	2280	Slope of southern exposure, xeromorphic
	3-10	−10				
	4-air	150		50.0607° N, 88.8100° E	2270	Foot of a north-facing slope, mesomorphic
	4-1	−1	2021–2022			
	4-10	−10				

**Table 4 insects-15-00064-t004:** Supercooling points (°C) of the darkling beetles.

Species, Stage	Season	*n*	Mean *SCP*	Min *SCP*
*Anatolica dashidorzsi*, Im	winter	35	−25.7 ± 0.9	−34.9
*Penthicus altaicus*, Im	autumn	18	−13.1 ± 0.9	−22.4
	winter	43	−25.0 ± 0.6	−31.9
*Bioramix picipes*, Im	winter	23	−21.7 ± 1.3	−29.4
*Bioramix picipes*, L	winter	7	−28.7 ± 0.3	−30.0
*Pedinus femoralis*, Im	winter	31	−19.3 ± 0.7	−26.1
*Blaps lethifera*, Im	winter	20	−15.1 ± 0.3	−17.4

**Table 5 insects-15-00064-t005:** Survival of adult darkling beetles when exposed to negative temperatures for a duration of 2 days.

Species	Date	*T*, °C	*n*	Survivors, %
*Anatolica dashidorzsi*	16.01	−5	19	63
	9.02	−5	14	71
	17.01	−20	22	36
	26.12	−24	33	6
*Penthicus altaicus*	16.01	−5	20	55
	9.02	−5	26	81
	17.01	−20	26	69
	26.12	−24	41	24
	25.01	−30	45	0
*Bioramix picipes*, Im	16.01	−5	18	95
	9.02	−5	16	100
	26.12	−22	29	55
	19.01	−25	17	18
	25.01	−30	30	0
*Blaps lethifera*, Im	16.01	−5	15	63
	9.02	−5	5	80
	17.01	−10	20	60
	9.02	−13	25	32
	29.12	−15	10	10
	1.02	−17	25	0

**Table 6 insects-15-00064-t006:** Lowest temperatures (and minimum of average daily temperature) in the soils and air in the study area during the winter seasons of 2021–2022 and 2022–2023.

Locality	Elevation, m a.s.l.	Logger No. *	2021–2022			2022–2023		
			1 cm	10 cm	Air	1 cm	10 cm	Air
Kosh-Agach	1761	1	−32.1 (−29.2)	−27.8 (−26.4)	–	−30.6 −26.6)	−26.7 (−24.4)	–
	1759	2	−20.6 (−20.1)	−18.6 (−18.4)	−41.2	–	–	–
Tabozhok	2280	3	−24.6 (−17.2)	−17.0 (−15.0)	–	−21.1 (−19.0)	−17.5 (−17.0)	–
	2270	4	−21.2 (−20.7)	−18.1 (−17.8)	−32.2	–	–	−35.2

* locations of the loggers, see Table 3.

**Table 7 insects-15-00064-t007:** Soil temperatures and snowpack depth in cold seasons 1989–2022 at the Kosh-Agach weather station [29].

Long-Term Average Minimum of Soil Temperatures ± SE, °C			Snow Depth in January, cm		
			At the Weather Station	Snow Survey	
20 cm	40 cm	80 cm		Mean	Maximum
−27.0 ± 0.45	−21.5 ± 0.35	−13.7 ± 0.33	8	5.8	15.3

## Data Availability

The data presented in this study are available in this article.

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
