# Peer review of "Insufficient Cold Resistance as a Possible Reason for the Absence of Darkling Beetles (Coleoptera, Tenebrionidae) in Pleistocene Sediments of Siberia"

_insects, 2024, doi:10.3390/insects15010064_

Round 1
Reviewer 1 Report
Comments and Suggestions for Authors
Line 117: Specify what you mean by different taxonomic groups. Are you referring to tribes, genera, species groups, or some combination of sub-family units?
Line 121: You state that 'three species of darkling beetles were collected.' There are two problems with this methodology. First, you need to test many more species than just three. As you state in the text, the darkling beetles are a huge family in the arid regions of Asia. I would suggest that you test at least ten species. Second, all of your specimens come from just two locations. To support your arguments, you should test beetles from many more sites, preferably in multiple regions. I appreciate that this will take a great deal of additional work, but as it stands now your conclusions are not sufficiently backed by robust data sets.
Please note: even though more work needs to be done, I am excited by the research project and strongly encourage the authors to pursue this project and manuscript.
Comments on the Quality of English LanguageThere is no reason to edit the English content at this point.
Author Response
Dear reviewer,
thank you for your acquaintance with the manuscript and your clearly expressed attitude towards it. We agree with your wishes, but they cannot improve the manuscript. Accepting them means abandoning what has been done and writing another work in the distant future.
As you know, the absence of darkling beetles in the Pleistocene of Northern Eurasia is a mystery, which may be due to both the specific conditions and the unknown nature of the defective physiology of the insects being studied. No one has ever made any attempts to find out the specific causes of the phenomenon. We took the first step and studied five species (not three, as you wrote). The result obtained, from our point of view, is promising and gives hope that other scientific teams will be included in the research.
Unfortunately, you do not even try to discuss the essence of the presented work, but express a complaint regarding the absence of what you would like to see. Yes, of course, 10 species are better than five; yes, the conclusions would be more reliable. But we have to start somewhere. We “paved the trail”, but you want us to immediately “build a highway”. However, there are compelling reasons (for example, such as limited funds and time limits for work on grants) that may prevent the extensive work you recommend from being carried out in the future.
Since we have not received specific claims or suggestions from you, our response is also limited to general considerations.
Reviewer 2 Report
Comments and Suggestions for Authors
The paper deals with cold resistance of five darkling beetle species from Altai (Chuya depression) that live in the coldest area still inhabited by tenebrionids despite strong temperature inversion. The aim is to explain why in Pleistocenic fossil faunas of periglacial steppes no or very few darkling beetles are found. The auhors demnstrate that even in the studied region the cold resistance of such tenebrionids is very low, because they overwinter only in particular microhabitats or under snow cover, they are freeze intolerant and die after 2 days at temperatures around 20-22 degrees. On this way the thesis is demonstrated and conclusions acceptable. I suggested some corrections on the PDF, paragraph 4.3.4. should be shortened because in my opinion a little verbose. In lines 632-636 a sentence is not clear to me, the auhors should check for correct reasoning.

The english is generally correct, I commented some arrors on the PDF text.
Author Response
Thank you very much for your positive assessment of our work, as well as for your comments and corrections of typos.
Reviewer: I suggested some corrections on the PDF, paragraph 4.3.4. should be shortened because in my opinion a little verbose.
Response: A sentence in lines 706-712 in paragraph 4.3.4 was changed:
The duration of depletion in numbers may depend on the depth of "temperature damage" of the population and the peculiarities of the species biology. The abundance may recover due to larvae hibernating in the soil stratum. The second source of population recovery at the bottom of the trough may be darkling beetles dispersing from lake depressions and from mountain slopes of southern exposure.
Reviewer: In lines 632-636 a sentence is not clear to me, the auhors should check for correct reasoning.
Response: A sentence in lines 632-636 was changed:
But the main problem probably lies in the literal overlap of available data. By default, it is assumed that beetles overwinter exactly where the loggers are installed, up to a depth of 10 cm; temperatures in deeper soil horizons were estimated using long-term average data from the Kosh-Agach weather station. With this method of assessing temperature conditions, successful overwintering of darkling beetles in the immediate surroundings of the weather station in the period from 1989 to 2022 would have been impossible: the average minimum soil temperatures to a depth of 50 cm were below the critical temperatures for darkling beetles (-22 °C) [29].
All other comments noted in the pdf file were taken into account and corrected.
Reviewer 3 Report
Comments and Suggestions for Authors
This paper investigates the cold resistance of five species of darkling beetles to elucidate why these beetles were absent from Pleistocene sediment fossils in Siberia. The authors found that the darkling beetles they tested are freeze intolerant. This topic is relevant in the field and aligns with the journal's scope. The experiments are well-designed, and the results are clearly presented. The introduction and discussion sections provide ample information. The references are appropriate. I recommend accepting this manuscript, subject to a few minor revisions:
Line 37: Delete "…"
Lines 119-127: Please provide a map showing the locations where all the darkling beetles were collected.
Table 1: Specify what "Im" and "L" stand for.
Figure 3 and Figure 4: The colors of dots used to represent specimens collected for the study and other localities are too similar, making it difficult to distinguish between them. Please consider changing the colors to improve visibility.
Table 2: Please add a description of the meaning of "+" and "-".
Line 231: Delete "…"
Lines 308-309: Ensure that all scientific names are italicized.
Line 311: Change "Figure 6" to "Table 4," since the statistical results were presented in Table 4.
Line 321: Add a space after "overwintered."
Author Response
Thank you very much for your high assessment of the work and for your comments
Line 37: Delete "…"
Response: The sentence has been rephrased and "…" has been deleted.
Lines 119-127: Please provide a map showing the locations where all the darkling beetles were collected.
Response: The map was provided.
Table 1: Specify what "Im" and "L" stand for.
Response: A note after the Table 1 was provided: «Im – imago; L – larvae».
Table 2: Please add a description of the meaning of "+" and "-".
Response: An explanation note after the Table 2 was added.
Line 231: Delete "…"
Response: "…" was changed to "from to".
Lines 308-309: Ensure that all scientific names are italicized.
Response: "Figure 6" and the caption to the figure has been deleted.
Line 311: Change "Figure 6" to "Table 4," since the statistical results were presented in Table 4.
Response: "Figure 6" was changed to "Table 4".
Line 321: Add a space after "overwintered."
Response: Done.
Round 2
Reviewer 1 Report
Comments and Suggestions for Authors
The revised manuscript is excellent and only requires minor editing of the English. I heartily recommend its publication, as it makes a very valuable contribution to Quaternary science and modern entomology.
Comments on the Quality of English LanguageOnly a few minor edits of the English are required, and can be carried out by the journal's copy-editing staff.